# Cell-Free DNA: Potential Application in COVID-19 Diagnostics and Management

**DOI:** 10.3390/v14020321

**Published:** 2022-02-04

**Authors:** Robert Stawski, Dariusz Nowak, Ewelina Perdas

**Affiliations:** 1Department of Clinical Physiology, Medical University of Lodz, 92-215 Lodz, Poland; dariusz.nowak@umed.lodz.pl; 2Department of Biostatistics and Translational Medicine, Medical University of Lodz, 92-215 Lodz, Poland

**Keywords:** COVID-19, cell-free DNA marker, SARS-CoV-2

## Abstract

WHO has declared COVID-19 as a worldwide, public health emergency. The elderly, pregnant women, and people with associated co-morbidities, including pulmonary disease, heart failure, diabetes, and cancer are the most predisposed population groups to infection. Cell-free DNA is a very commonly applied marker, which is elevated in various pathological conditions. However, it has a much higher sensitivity than standard biochemical markers. cfDNA appears to be an effective marker of COVID-19 complications, and also serves as a marker of certain underlying health conditions and risk factors of severe illness during COVID-19 infection. We aimed to present the possible mechanisms and sources of cfDNA released during moderate and severe infections. Moreover, we attempt to verify how efficiently cfDNA increase could be applied in COVID-19 risk assessment and how it corresponds with epidemiological data.

## 1. Introduction

Severe acute respiratory syndrome coronavirus 2 (SARS-CoV-2) has caused the greatest worldwide pandemic of the 21st century. The consequences of coronavirus disease 2019 (COVID-19) have been significant for the economy, social life, and public health worldwide. Most patients do not demonstrate symptoms, or are mildly symptomatic, but some people infected with SARS-CoV-2 experience extensive inflammation and severe homeostasis imbalance. A severe infection first begins in the respiratory system; however, this might be followed by septic shock and multiple organ failure [1]. 

Cell-free DNA (cfDNA) has been studied extensively for last few decades; since then, almost every aspect of the structure of cfDNA has been studied. Analysis of DNA integrity allows to determine the process responsible for the release of cfDNA. Necrosis or NETosis disintegrate a membrane and release random long fragments (<10,000 bp), whereas apoptosis is preceded by the shrinkage of the cell, fragmentation into membrane-bound apoptotic bodies, and the release of 180–200 bp fragments of DNA (Figure 1) [2]. The most common method used for the quantification of cfDNA isolated from body fluids is quantitative real-time PCR (qPCR), based on TaqMan gene expression assay. Of note, cfDNA extraction is also possible from unpurified plasma [3], and using direct qPCR, which seems to be a sufficiently sensitive procedure for the quantification of cfDNA concentrations, might find broad applicability. The detection precision is rising and this is followed by new, more sensitive methods, such as ddPCR [3].

Currently, during COVID-19 infection, patient assessment is based on standard diagnostic markers, such as CRP, creatine, procalcitonin, or cytokines. However, cfDNA seems to be tremendously more sensitive compared to them [4]; moreover, fast kinetics within minutes allows much better monitoring comparing to CRP (with half-life of 19 h) [5]. As mentioned above, circulating cfDNA can increase via numerous mechanisms and in response to a variety of pathophysiological factors. This suggests its limited specificity as a biomarker of complications of COVID-19 infection. However, CRP also has a relatively low specificity, but is commonly used to evaluate the severity of inflammatory responses to various factors, including bacterial, viral, fungal infection, sepsis, septic shock, or trauma. Serum CRP is also elevated in patients with cancer. Similarly, procalcitonin, which is useful marker of bacteremia, and sepsis could be elevated in pediatric patients with immunological disorders [6]. In addition, other non-infectious causes of systemic inflammation (e.g., trauma, surgery, burn injury, chronic kidney disease) have been reported to increase circulating procalcitonin [7,8]. Therefore, cfDNA could serve as an additional biomarker of serious complications that threaten the lives of patients infected with COVID-19.

Substantial evidence indicates that angiotensin-converting enzyme 2 (ACE2) is the key factor in COVID-19 infection (Figure 1). Physiologically, ACE2 cleaves the angiotensin I hormone into vasoconstricting angiotensin II. However, ACE2 also serves as an entry point into cells for some coronaviruses, including SARS-CoV-2 [9]. ACE2 is an enzyme attached to the cell membranes of cells located in the lungs (lung type II alveolar cells), arteries and veins (endothelial cells), intestines (enterocytes of the small intestine), and arterial smooth muscle cells, and in most organs (e.g., heart and kidney) [10]. 

The binding of SARS-CoV-2 to host cell ACE2 may dysregulate erythropoiesis through the downstream angiotensin II pathway. Moreover, it was found that angiotensin II regulates normal erythropoiesis and promotes early erythroid proliferation through unclear downstream mechanisms [11,12,13]. Lui et al. (2002) suggested that cfDNA originates from hematopoietic cells [14]. Therefore, a significant increase in cfDNA derived from red blood progenitor cells may be caused by injury to red cell precursors through direct or indirect processes [15,16].

cfDNA plays an essential role in immune system homeostasis. Cells are treated with native plasma expressed genes, whose products maintain immune system homeostasis. The cells treated with plasma after DNAse directly elevate production of mRNA for interleukin 8. These also upregulated complement compounds at the proteomic level [17]. The molecular mechanism of cfDNA is similar to that of other damage associated molecular patterns (DAMPs), which can stimulate proinflammatory activity though the TLR9 receptor. The principal cytosolic DNA pathway seems to be stimulator of interferon genes (STING), which activates TANK binding kinase 1 (TBK1) and IFN [18]. In addition, proinflammatory cfDNA, either in the presence or absence of histones, has been shown to modulate several procoagulant pathways by stimulating thrombosis or inhibiting fibrinolytic activity [19,20].

Recently, various types of cfDNA-based therapies have been tested; for instance, recombinant human DNase (rhDNase) has been used in sepsis, but also in COVID-19 [21]. Furthermore, nucleic acid-binding nanoparticles (NABN) and polymers (NABPs) have been successfully applied in sepsis or influenza infection [22,23]. Of note, all experiments were characterized with positive therapy effects. 

NETosis is a unique form of immune cell death that is characterized by a release of decondensed chromatin into the extracellular space to catch a pathogen. cfDNA and other DAMP molecules, such as citrullinated histones H3 and myeloperoxidase, are major structural elements of NETs. Numerous reports have suggested a fundamental role of NETosis in COVID-19 infection [24,25,26,27]. Hardly any cell-free DNA molecules can be detected in the blood of healthy individuals. However, in the case of severe tissue or organ damage, the blood might be full of released DNA long before the spread of infection. Nevertheless, under normal physiological conditions, even if dying cells release their remnants, they are efficiently removed. Structural compartments are cleared mainly by the liver, whereas cfDNA is rapidly degraded by endonucleases. Thus, under physiological conditions, cellular remnants or cfDNA are normally not inflammatory due to their fast degradation.

Due to persistent excretion or inefficient clearance, circulating cfDNA exhibit a negative effect on body hemostasis. In this manuscript, we raise the question of if it might be an important element of COVID-19 pathogenesis. Currently, numerous reports present the application of cell-free DNA as a marker in many age-related and pathological conditions, such as cancer, diabetes, sepsis, aseptic inflammation, as well as in transplantations [28]. Measurements of cell-free DNA in serum or plasma are minimally invasive and highly precise diagnostic methods, providing real-time observation for a wide spectrum of pathologies, including COVID-19. cfDNA detection as a diagnostic method can be applied relatively easily, even in unpurified blood [3]. 

The aim of the present review is to verify how efficiently cfDNA increase could be applied to COVID-19 risk assessment, and how it corresponds with epidemiological data. Furthermore, we tried to explain the reasons for the increase in cfDNA during the course of COVID-19 infection, starting from moderate infection, and ending with advanced sepsis and multi-organ failure.

## 2. cfDNA and Risk of COVID-19

### 2.1. Age, Obesity and Diabetes 

There is a long list of conditions that might increase the risk for severe illness from COVID-19. According to the World Health Organization (WHO), COVID-19 mortality is strongly related to patient age and obesity. The mortality risk rises significantly with age, especially after the age of sixty. At the ages of 0–20 or 20–30, the mortality rate is 0.2%, whereas over 80 years of age, it can be as high as 13–14%. Moreover, patients with obesity and diabetes are also at risk of severe COVID-19. Lighter et al. demonstrated that patients with COVID-19 and a BMI between 30 and 34.9 were two times more likely to be admitted to a critical care unit than patients without obesity [29]. Three meta-analyses confirmed that obesity could increase the risk of infection and poor outcome in patients with COVID-19 [30,31,32]. However, obesity status for cancer patients was not associated with mortality [33]. Al Sabath et al. (2020) suggested that patients with both obesity and diabetes must be categorized as a high-risk group [34]. This was confirmed by two independent meta-analyses [35,36].

There is more and more evidence that cell-free DNA could serve as a marker of human aging or obesity [37,38,39,40]. Increased levels of body fat may increase levels of proinflammatory cytokines, resulting in a state of chronic inflammation or oxidative stress, leading to the processes of apoptosis and necrosis. MPO-DNA complexes, one of the markers of NETosis, was found to be at a high level in a group of patients with obesity who underwent sleeve gastrectomy compared to healthy controls. Interestingly, patients with reduced MPO-DNA complexes after surgery presented a reduced body weight and BMI and an improved glycemic status. On the other hand, a sub-group with persisting high levels of MPO-DNA complexes after surgical treatment had a history of stroke and thromboembolism, and, therefore, may represent a high CV risk population. This may suggest that only weight loss may not modify neutrophil activation [41]. Moreover, evidence indicates an association of NET formation in the pathophysiology and complication of diabetes [42,43,44]. In addition, it is well known that, after forty years of age, the risk of multiple disorders, including cardiovascular disease, hypertension, or diabetes, increases. Moreover, age-associated accumulation of metabolites or cell debris might be linked with chronic systemic inflammation.

### 2.2. Vitamin D Deficiency

Hypovitaminosis D, as well as diabetes, cardiovascular events, and associated comorbidities, are the main causes of severe clinical complications in COVID-19 patients. However, the effect of Vitamin D on the severity and outcome of COVID-19 has not yet been fully recognized. Many recent reports indicate a prevention or treatment effect of vitamin D on the course of COVID-19 [45,46,47]. Vitamin D deficiency has been described as a risk factor in the development of inflammatory processes, such as acute lung injury, acute respiratory distress syndrome (ARDS), and infectious diseases, such as respiratory tract infections. Bearing in mind the fact that vitamin D plays an important function in immunity, its supplementation might enhance the immune system of COVID-19 patients and reduce the severity of the disease in vitamin D-deficient individuals through modulation of the innate and adaptive immune systems [48].

Interestingly, vitamin D treatment significantly reduces the level of cell-free DNA, proinflammatory factors, and NETosis level [49,50]. The combined blood vitamin D status and cell-free DNA level might provide useful information regarding the clinical course, the extent of lung involvement, and outcome of patients with COVID-19 [51]. However, a recent meta-analysis shows contradictory reports. Borsche et al. recommend raising serum 25(OH)D levels [52]; on the other hand, Chen et al. claimed that low vitamin D levels do not aggravate COVID-19 risk or death, and that vitamin D supplementation does not improve outcomes in hospitalized patients with COVID-19 [53].

### 2.3. Cancer

According to WHO reports, some cancer patients might be at increased risk of serious illness from COVID-19 infection. This is associated with the general weakening of the body and immune system caused by the disease and therapy. On the other hand, cancer patients benefit from taking certain medications. Reports suggested that dexamethasone and tocilizumab may be beneficial in patients who receive either oxygen or mechanical ventilation due to COVID-19. Tocilizumab is an anti-IL-6 receptor antibody that inhibits the binding of IL-6 to IL-6 receptors, blocking IL-6 signaling and reducing inflammation, which limits the development of a hypercytokinemia, also called a “cytokine storm”. This is a physiological reaction characterized by a sudden release of cytokines in large quantities, which might cause multisystem organ failure and even death (more in Section 3.1) [54,55]. A higher incidence of COVID-19 with more severe symptoms has been noted in patients with lung cancer [56,57,58]. Angiotensin-converting enzyme 2 (ACE2), being the only experimentally established SARS-CoV-2 receptor, could assist the virus in entering cells and its expression level is considered to indicate predisposition to COVID-19. Elevated ACE2 expression was found in both lung tumors in non-small cell lung cancer (NSCLC), including lung adenocarcinoma (LUAD), and lung squamous cell carcinoma (LUSC), compared to normal tissues. This could explain why SARS-CoV-2 infection more frequently affects the respiratory system than other body parts [59,60,61]. However, elevated ACE2 expression was statistically related to a shortened overall survival rate in LUAD and a significantly longer disease-free survival in LUSC, which implies a very complex connection between ACE2 and lung cancer, and the role of ACE2 expression [61].

There are currently hundreds of reports that confirm an altered level of cell-free DNA in different types of cancer [62]. The release of cfDNA into body fluids is a result of cancer development by malignant tissues, and also by surrounding tissues suffering from starvation, hypoxia, or other factors associated with cell death. In fact, the use of cfDNA is potentially a minimally-invasive alternative option to biopsy for diagnosis and could also be a tool for prognosis and predictive evaluations. Moreover, cfDNA helps to distinguish between malignant and benign neoplasia, as well as controls, tumor type, grade, lymph node status, primary tumor, and metastases of many organs, glands, and tissues, including breast [63,64,65], prostate [66], colon [67], liver [68], ovary [69,70], endometrium [71], thyroid [72], and lung [73].

### 2.4. Autoimmune Disease

In the literature, subjects with autoimmune diseases have been found to be at higher risk of death from infections, and at higher risk of nonfatal infections compared to the general population [74]. An increased risk of hospitalization or serious infections have been reported in subjects with rheumatoid arthritis (RA) and systemic lupus erythematosus (SLE) [74].

Notably, it is not well known how coronavirus affects people with autoimmune diseases, or those who take drugs that influence the immune system. However, in general, drugs that treat autoimmune diseases, such as biologics and corticosteroids, may contribute to a higher risk of severe viral infection [75]. As with cancer patients, RA patients taking tocilizumab also benefit during COVID-19 infection [55].

It is worth pointing out that both SLE and RA are characterized by a significantly increased level of cell-free DNA [76,77]. In SLE, cfDNA was increased by four times, whereas in RA it was increased by three times [78]. The association of cfDNA levels with serological parameters in both diseases, e.g., anti-dsDNA in SLE and RA, suggests that cfDNA reflects common processes involved in both diseases, including inflammation and cell death [76]. 

### 2.5. Recipients for Organ Transplantation

COVID-19 appears to put patients with cardiovascular diseases, as well as those on immunosuppressive medication due to organ transplantation, at risk. A noninvasive strategy, involving the use of measurements of donor-derived cell-free DNA (dd-cfDNA), is employed in order to prevent acute rejection in heart and kidney transplant recipients. Plasma dd-cfDNA has shown a high negative predictive value for acute rejection, but it might also be equally effective in identification of other forms of cardiac injury, such as vasculopathy. In the context of COVID-19, noninvasive monitoring of rejection is advantageous as it allows to minimize a patient’s contact with the healthcare system. An increase in dd-cfDNA in a heart transplant patient suggests subclinical allograft damage caused by viral infection. Patients receiving immunosuppressive therapy may persistently appear to be virus-positive. Thus, it is difficult to make recommendations regarding the length of a patient’s self-quarantine and the timing required to make personal appointments with a cardiologist to undergo tests [79].

In kidney transplantation procedures, the dd-cfDNA test for screening for rejection, as well as clinical information, can enable to determine whether it is necessary for a transplant recipient to visit a medical facility [80].

### 2.6. Respiratory System Diseases

The most common lung disease, other than cancer, is chronic obstructive pulmonary disease (COPD), which increases the risk of severe illness associated with COVID-19 [81]. Main outcomes show that the prevalence of COPD in COVID-19 patients was low, but that the risk of severity (63%) and mortality (60%) were high [82].

Smoking is most likely associated with progression and unfavorable outcomes of COVID-19 [83]. Current smokers demonstrate increased gene expression of ACE2 than former smokers and non-smokers [84,85]. There is further evidence that ACE2 expression is closely related to nicotine exposure [86,87]. Hence, it can be concluded that smoking affects ACE2 expression and consequently is a risk factor for COVID-19. Moreover, some studies suggest that, upon admission to a hospital, circulating cfDNA level may serve as an effective tool for early diagnosis of smoke inhalation injury [88]. Patients demonstrated elevated cfDNA levels and the levels correlated with hospitalization time. Cell-free DNA appears to be a potentially valuable marker for severity and follow-up in patients with smoke inhalation injuries. Cell-free DNA also correlated with CO intoxication levels and daily cfDNA measurements reflected the recovery of hospitalized patients.

All outcomes related to admission, testing, screening, ventilation, recovery, and death need to be evaluated in relation to smoking status and adjusted to comorbid conditions, such as COPD. Leung et al. [89] demonstrated, in three separate cohorts with gene expression profiles from bronchial epithelial cells, that ACE-2 expression was significantly elevated in COPD patients compared to control subjects. This evidence implies that COPD patients display the machinery required for SARS-CoV-2 cellular entry. Plasma cfDNA might offer a novel technique to identify COPD patients at increased risk of poor outcomes. In COPD, cell-free DNA increases by more than four times, thus plasma cfDNA might offer a novel technique to identify COPD patients at increased risk of poor outcomes [90].

## 3. cfDNA and COVID-19 Complications

The latest reports show that cfDNA levels positively correlated with the severity of COVID-19 disease and confirm that the cfDNA profile noted upon admission allowed to identify patients who later required intensive care or died during hospitalization (Figure 2). The increase shown in this figure is an effect of the cumulative release of cfDNA from different sources, depending on the actual location of the infection (lungs, blood immune cells). The suggested scale of cfDNA growth is presented in Figure 3. Andargie et al. demonstrated that the kidney, heart, lung, hematopoietic cells, vascular endothelium, hepatocytes and adipocytes are the main sources of cfDNA in COVID-19 [91]. 

### 3.1. Blood and Immune System

There is more and more evidence showing that accelerated progression of the COVID-19 disease is linked to excessive inflammation, called a “cytokine storm”, which causes major systemic perturbations. This condition manifests as high fever, swelling, extreme tiredness, and nausea. In some cases, the immune reaction can even result in death. A cytokine storm is an early phase of sepsis, described by excessive inflammation [94]. Apart from proinflammatory cytokines, COVID-19 seems to exhibit a dysregulated immune response, characterized by sustained reduction of the peripheral lymphocyte counts, known as lymphopenia. Moreover, the degree of lymphopenia has been shown to correlate with the severity of COVID-19 [61]. Therefore, viral sepsis is a clinical manifestation of severe or critically ill COVID-19 patients. Understanding the mechanism of viral sepsis in COVID-19 will provide these patients with better clinical care [95].

cfDNA is a well-established stress marker in many pathologies, including sepsis. The inflammatory and oxidative stress caused by sepsis may increase cell apoptosis/necrosis, and, as a consequence, increase many markers, as well as cell-free DNA levels. Patients with sepsis have twenty-two times higher levels of cfDNA compared to non-septic patients. Moreover, it was found that the level of cell-free DNA allows to categorize sepsis patients admitted to emergency rooms into survivors and non-survivors [96,97]. 

Numerous reports suggest a fundamental role of NETosis in COVID-19 infection [24,25,26,27]. Virus-induced NETs can circulate in the blood in an uncontrolled way, leading to an extreme systemic response by the body, followed by the production of immune complexes and chemokines, finally increasing inflammation [26].

cfDNA, apart from citrullinated histones H3 and myeloperoxidase, is a major structural element of NETs. Huckriede et al. observed increased H3 and cfDNA levels in critically ill COVID-19 patients. They indicated the severity of a cellular injury. Moreover, the increase in neutrophil counts shows a significant role of neutrophil response and the process of NETosis in the disease [98,99].

The pathological effect of cfDNA (besides being proinflammatory) involves its ability to trigger blood coagulation, as well as to inhibit clot lysis, which may lead to COVID-19 pathogenesis. This is done primarily through provoking macrovascular and microvascular thrombosis [100,101,102,103]. Damage to endothelial cells may contribute to occurrence of COVID-19 coagulopathy and the prothrombic state [104].

In COVID-19 infection, pulmonary thrombosis appears to be a common consequence of pneumonia and some clinicians recommend implementing anticoagulation therapy (rather than prophylactic dosing) as routine management of patients with COVID-19, believing it will be beneficial in preventing microvascular thrombosis [105].

cfDNA and other DAMPs molecules may have harmful effects on a host. An elevated level of cfDNA appears to have a prognostic value in predicting the poor outcome in pathological conditions, characterized by excessive activation of coagulation and inflammation. Pharmacological strategies that inhibit NETosis or those which neutralize toxic effects of cfDNA are a focus of attention for clinicians. It is possible that a combination of therapies that reduce coagulation and inflammation will appear to be most beneficial [103].

### 3.2. Multiorgan Failure

The presence of coronavirus has been confirmed in the heart, liver, and blood of many patients [106,107]. It is still unknown if COVID-19 directly targets these organs or if they are damaged by extensive inflammation. A significant severe viral renal infection in some patients could explain the increased risk of acute kidney injury in patients with COVID-19. Cardiovascular complications occur frequently and are associated with poor prognosis. Notably, of 100 COVID-19 patients who recovered from the disease, 78% had confirmed cardiac problems and 60% had ongoing myocardial inflammation [108]. In addition, SARS-CoV-2 was self-diagnosed in over 60% of patients [109]. cfDNA was higher in diabetic patients with vascular complications in comparison to controls [110]. In addition, cfDNA may also be used to assess allograft rejection and injury [111]. The mean concentration of cfDNA in patients with acute myocardial injury was 5-fold higher during the onset of disease compared with healthy volunteers [112]. In ischemic heart disease, cfDNA increases up to 50-fold compared with healthy controls [113,114].

General guidance published by the WHO does not provide information regarding COVID-19 risks in patients with previously diagnosed thyroid issues. Moreover, there is no information on whether patients with COVID-19 (symptomatic or asymptomatic), who have not previously had thyroid problems, develop an endocrine thyroid dysfunction after COVID-19 infection. Recent evidence shows that patients affected by COVID-19 and demonstrating more severe symptoms have lower serum levels of fT3 and TSH compared with controls. This may reflect direct damage to the thyroid or even pituitary gland by the virus [115,116]. Moreover, there is a great deal of evidence of changes in cfDNA in patients with a thyroid dysfunction, including thyroiditis, both benign and malignant [72,117].

Hepatic involvement in COVID-19 could be related to the direct cytopathic effect of the virus, an uncontrolled immune reaction, sepsis, or drug-induced liver injury. In the current pandemic, hepatic dysfunction has been observed in 14–53% of COVID-19 patients, particularly in those with a severe course of disease. Cases of acute liver injury have been reported and contribute to a higher mortality [118]. Thus, cfDNA seems to be an efficient marker that can be applied, not only to hepatic problems, but also to all gastrointestinal disorders [119]. 

Many classical biochemical markers clearly reflect organ damage. However, none of them can be efficiently applied to a number of organs simultaneously.

As shown in Figure 3, cell-free DNA increase has been observed in almost all possible COVID-19 complications. The increase ranges from 2-fold under psychosocial stress conditions [120], 4-fold in chronic kidney disease [121] or lung disease, and up to 5–6-fold in cardiovascular disease (even more in sepsis) [122,123,124].

**Figure 3 viruses-14-00321-f003:**
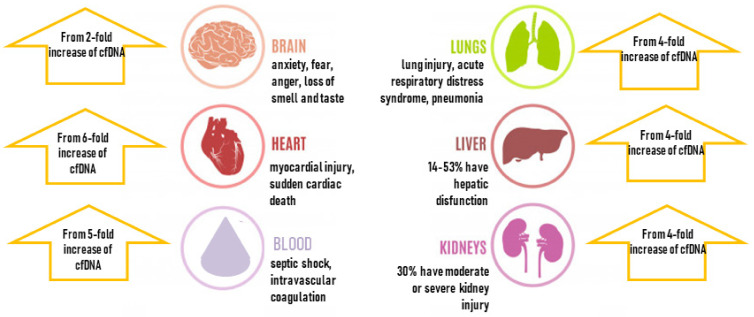
Scheme showing the most common complications of COVID-19 in the context of cell-free DNA fold changes [4,24,106,107,120,121,122,123,124,125].

## 4. Conclusions

cfDNA secretion in the course of COVID-19 infection might be associated with many factors. Firstly, activated immune cells release significant amounts of circulating molecules, including cfDNA. Secondly, infection with SARS-CoV-2 has been also shown to cause lung complications, such as pneumonia or acute respiratory distress syndrome, in consequence leading to an abnormally low level of oxygen in the blood. This leads to accumulation of characteristic cfDNA indicators, such as free radicals, changes in pH, lactic acid, and electrolytes. These processes, in consequence, cause cellular damage and death, leading to a release of cellular compartments, including nucleic acids. Lastly, wide distribution of ACE2 receptors allows a virus to experience multiorgan spread and extensive disease distribution. However, a direct relationship between COVID-19 and cfDNA cannot be proven. However, cfDNA appears to be an effective marker of COVID-19 complications, and also serves as a marker of certain underlying health conditions and risk factors of severe illness during COVID-19 infection. 

Effective monitoring of factors associated with COVID-19 mortality can help to recognize patients who are at higher risk of a poor prognosis. Good markers can provide an early warning to initiate and facilitate appropriate interventions [125,126]. To sum up, cell-free DNA is a marker with a wide spectrum of applications, successfully applied for many different diseases. Moreover, it is also characterized by a much higher sensitivity than standard biochemical markers [4,127]. To this end, cfDNA tests can be greatly improved by adding a combination with several standard diagnostic biochemical biomarkers to them.

## Figures and Tables

**Figure 1 viruses-14-00321-f001:**
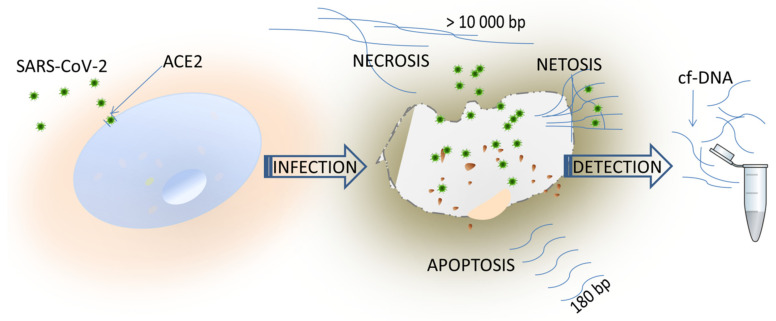
ACE2 as the entry receptor for SARS-CoV-2 and possible sources of cfDNA during COVID-19 infection.

**Figure 2 viruses-14-00321-f002:**
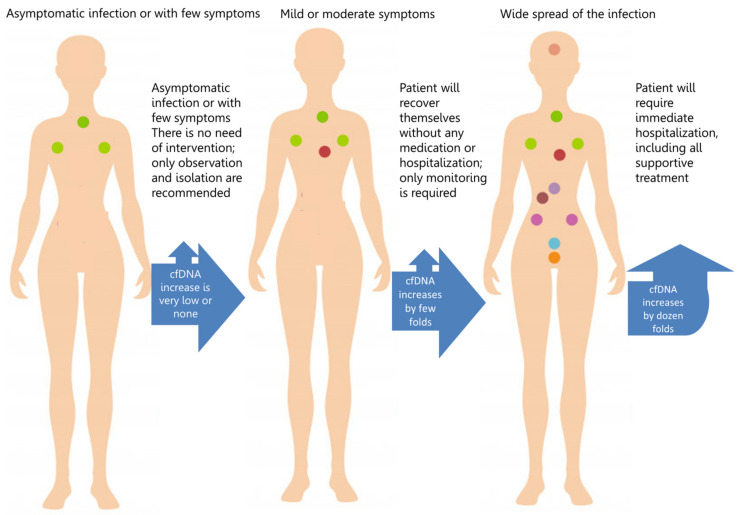
COVID-19 management based on cell-free DNA analysis [92,93]. Schematic representation of how to deal with a patient, based on the level of cfDNA, reflecting the current condition and the stage of disease in the COVID-19 patient. Filled spots represent the potential cfDNA releasing organ (details shown in Figure 3).

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
