# Peer review of "Cell-Free DNA: Potential Application in COVID-19 Diagnostics and Management"

_viruses, 2022, doi:10.3390/v14020321_

Round 1

Reviewer 1 Report

This is a sound and comprehensive review. I found only 2 minor issues that should be addressed by the Authors.

  1. Reference no. 71 (preprint) should be replaced by an official version published version (doi: 10.1038/s41598-021-95209-x.). If preprint at Researgate and publication in Sci. Rep. differ substantially, then both should be cited.
  2. Figure 2 states that patients with a several-fold cfDNA increase suffer only from mild symptoms, whereas fig. 3 lists the high percentage of severe symptoms in patients with 2-6 - fold cfDNA increase. Could these figures be reconciled?

Author Response

This is a sound and comprehensive review. I found only 2 minor issues that should be addressed by the Authors.

  1. Reference no. 71 (preprint) should be replaced by an official version published version (doi: 10.1038/s41598-021-95209-x.). If preprint at Researgate and publication in Sci. Rep. differ substantially, then both should be cited.
  2. Figure 2 states that patients with a several-fold cfDNA increase suffer only from mild symptoms, whereas fig. 3 lists the high percentage of severe symptoms in patients with 2-6 - fold cfDNA increase. Could these figures be reconciled?

Thank you very much for those constructive comments. They significantly improved the quality of our manuscript. According to them, we made the following changes in our manuscript:

Response 1 : 

Both versions are included in manuscript (ref. no [99] & [100]).

Response 2:

The figures are complementary to each other however are not directly linked this is because the third figure shows the increase in various diseases, while the second figure presents the increase due to a cumulative effect on the individual organs/systems (in example from blood and lung). To clarify this we have added the text to the main text.

Thus in line 265-273  “The latest report shows that cfDNA levels positively correlated with severity of the COVID-19 disease and confirms that the cfDNA profile noted upon admission enabled to identify patients who later required intensive care or died during hospitalization (Fig. 2). The increase shown in this figure is an effect of the cumulative release of cfDNA from different sources depending on the actual location of the infection (lungs, blood immune cells). Whereas the suggested scale of cfDNA growth has been presented in figure 3. Andargie et al. demonstrate that the kidney, heart, lung, hematopoietic cells, vascular endothelium, hepatocytes, and adipocytes are the main source of cfDNA in COVID-19 [91]”

Reviewer 2 Report

While the current review talks about a very relevant topic, i.e., potential of using cell-free DNA as a biomarker to study progression of COVID-19, the current form of this paper does not provide any critical review of the existing literature. While reading this review it appears that the cfDNA increases in the body in response to every disease condition, be it cancer, inflammation, lung injury etc. Under such a circumstace it is not clear how it could make a good case to be used as a biomarker for COVID-19. Also certainly release of cell free DNA occures due to cell death or during ageing but the authors need to provide a more in-depth assessment of its role in the cellular signalling and discuss the pros and cons of its application in COVID-19 research. It is mentioned loosely in places that methods to regulate the release of cfDNA could be used as therapy but it is not clear how? A more in depth and critical review of the existing literature is required to make a strong case for this review. Also the authors need to define the purpose of this review in the Introduction. In the current form each paragraph has an independent chain of thought without any cohesive link to the overall purpose of this review.

Author Response

While the current review talks about a very relevant topic, i.e., potential of using cell-free DNA as a biomarker to study progression of COVID-19, the current form of this paper does not provide any critical review of the existing literature. While reading this review it appears that the cfDNA increases in the body in response to every disease condition, be it cancer, inflammation, lung injury etc. Under such a circumstace it is not clear how it could make a good case to be used as a biomarker for COVID-19. Also certainly release of cell free DNA occures due to cell death or during ageing but the authors need to provide a more in-depth assessment of its role in the cellular signalling and discuss the pros and cons of its application in COVID-19 research. It is mentioned loosely in places that methods to regulate the release of cfDNA could be used as therapy but it is not clear how? A more in depth and critical review of the existing literature is required to make a strong case for this review. Also the authors need to define the purpose of this review in the Introduction. In the current form each paragraph has an independent chain of thought without any cohesive link to the overall purpose of this review.

Response:

Thank you for challenging, however voluble reviews. Some responses have been integrated with responses to other reviewers due to the convergence of these comments. Changes listed below:

  • …the current form of this paper does not provide any critical review of the existing literature.

Re: To make this review more comprehensive and critical we have added several meta-analysis. Moreover, additional comments and literature have been added:

Lines 130-133  “Three meta-analysis confirm that obesity could increase the risk of infection and poor outcome in patients with COVID-19 [30-32]. However, obesity status for cancer patients were not associated with mortality [33] “

Lines 134-135 ”This is confirmed by two independent meta-analyzes [35,36]”

In section 2.3, lines 175-180 “On the other hand, cancer patients profit from taking certain medications. Reports suggested that dexamethasone and tocilizumab may be beneficial in patients who receive either oxygen or mechanical ventilation due to COVID-19. Tocilizumab is an anti-IL-6 receptor antibody that inhibits the binding of IL-6 to IL-6 receptors, blocking IL-6 signaling and reducing inflammation [54,55].”

In section 2.4,  lines 211-212 “However, as with cancer patients, RA patients taking tocilizumab also benefit during COVID-19 infection [55].”

  • While reading this review it appears that the cfDNA increases in the body in response to every disease condition, be it cancer, inflammation, lung injury etc. Under such a circumstace it is not clear how it could make a good case to be used as a biomarker for COVID-19.

We agree that cfDNA increases due to many different factors but regardless of the condition (cancer, age, obesity etc), cfDNA measurement can quickly provide information about the COVID-19 patient's condition.

Lines 41-55 “Currently during COVID-19 infection patient assessment is based on standard diagnostic markers such as CRP, creatine, procalcitonin, or cytokines. However, cfDNA seems to be tremendously more sensitive comparing to them [4], moreover fast kinetics within minutes allows much better monitoring comparing to CRP (with half-life of 19 hours) [5]. As was mentioned above the circulating cfDNA could increase via various mechanisms and in response to a variety of pathophysiological factors. This suggests its limited specificity as a biomarker of COVID-19 infection complications. However, CRP has also relatively low specificity but is commonly used to evaluate the severity of inflammatory response to various factors including bacterial, viral, fungal infection, sepsis, septic shock ,or trauma. Serum CRP is also elevated in patients with cancer. Similarly, procalcitonin that is useful marker of bacteremia and sepsis could be elevated in pediatric patients with immunological disorders [6]. Also, other non-infectious causes of systemic inflammation (e.g. trauma, surgery, burn injury, chronic kidney disease) were reported to increase circulating procalcitonin [7,8]. Therefore, cfDNA could serve as an additional biomarker of serious complications that threaten life of patients infected with COVID-19”

 (3) Also certainly release of cell-free DNA occures due to cell death or during ageing but the authors need to provide a more in-depth assessment of its role in the cellular signalling and discuss the pros and cons of its application in COVID-19 research.

Lines 70-80 “cfDNA plays an essential role in the immune system homeostasis. The cells treated with the native plasma expressed genes whose products maintain immune system homeostasis. Whereas, the cells treated with identical plasma samples however treated with DNAsis directly elevated production of mRNA for interleukin 8. They also upregulated the complement compounds at the proteomic level [17]. cfDNA belongs to DAMPs (Dam-age Associated Molecular Patterns) which and though receptor TLR9 can stimulate proinflammatory activity. Whereas the principal cytosolic DNA sensor seems to be STING (Stimulator of Interferon Genes) which activates TANK binding kinase 1 (TBK1) and IFN [18]. Besides proinflammatory cfDNA, either in the presence or absence of histones, has been shown to modulate several procoagulant pathways by stimulating thrombosis or inhibits fibrinolytic activity [19,20].”

  • It is mentioned loosely in places that methods to regulate the release of cfDNA could be used as therapy but it is not clear how?

Re: Introduction: Lines 81-89 ” Recently, various types of cfDNA based therapies have been tested. For instance, recombinant human DNase (rhDNase) in sepsis but also in COVID-19 [21]. Furthermore, nucleic acid-binding nanoparticles (NABN) and polymers (NABPs) have been successfully applied in sepsis or influenza infection [22,23]. Notable, all experiments were characterized with positive effects of therapy. The aim of the presented review was to verify how efficiently cfDNA increase could be applied in COVID-19 risk assessment and how it corresponds with epidemiological data. Furthermore, we tried to explain the reasons for the increase in cfDNA during the course of COVID-19 infection, starting from moderate infection, and ending with advanced sepsis and multi-organ failure.”

  • A more in depth and critical review of the existing literature is required to make a strong case for this review. Also the authors need to define the purpose of this review in the Introduction. In the current form each paragraph has an independent chain of thought without any cohesive link to the overall purpose of this review.

Purpose has been clarified in the Introduction in lines 111-115 “The aim of the presented review was to verify how efficiently cfDNA increase could be applied in COVID-19 risk assessment and how it corresponds with epidemiological data. Furthermore, we tried to explain the reasons for the increase in cfDNA during the course of COVID-19 infection, starting from moderate infection, and ending with advanced sepsis and multi-organ failure.”

Re: To clarify and justify purpose we change the abstract in lines 14-17 “We aimed to present the possible mechanisms and sources of cfDNA released during moderate and severe infection. Moreover, we attempt to verify how efficiently cfDNA increase could be applied in COVID-19 risk assessment and how it corresponds with epidemiological data”

Reviewer 3 Report

In this paper, R. Stawski et al. discussed the usefulness of cell-free DNA testing in different diseases with a particular interest in COVID-19. The paper is well written and provides complex information on the biology and possible application of cell-free DNA testing. The hypothesis of cell-free DNA testing in COVID-19 patients was adequately discussed. The only is to consider providing short information on cell-free DNA testing.

Author Response

In this paper, R. Stawski et al. discussed the usefulness of cell-free DNA testing in different diseases with a particular interest in COVID-19. The paper is well written and provides complex information on the biology and possible application of cell-free DNA testing. The hypothesis of cell-free DNA testing in COVID-19 patients was adequately discussed. The only is to consider providing short information on cell-free DNA testing

Response:

Thank you very much for these constructive comments and suggestions. They significantly improved the quality of our manuscript. According to them, we made the following changes in our manuscript:

(1) The hypothesis of cell-free DNA testing in COVID-19 patients was adequately discussed. The only is to consider providing short information on cell-free DNA testing

Re: (1) In lines 29-40 “Cell-free DNA (cfDNA) is extensively studied from last few decades, since then almost every aspect of the structure of the cfDNA has been studied. Analysis of DNA integrity allows to determine the process responsible for the release of cfDNA. Necrosis or NETosis disintegrate the membrane and release random long fragments (<10 000 bp) whereas apoptosis is preceded by shrinkage of the cell, fragmentation into membrane-bound apoptotic bodies, and release of 180-200 bp fragments of DNA (Figure 1) [2].”

Moreover in lines 34-40 “ The most common method used for quantification of cfDNA isolated from body fluids is quantitative real-time PCR (qPCR) based on TaqMan gene expression assay. Notable, cfDNA extraction is also possible from unpurified plasma [3], and using direct qPCR, which seems to be sufficiently sensitive procedure for the quantification of cfDNA concentrations might find broad applicability. The precision of detection is rising and is followed with a new more sensitive method such as ddPCR [3].”

Reviewer 4 Report

The authors have reviewed the potential application of cfDNA for COVID-19 diagnostics and management. Role of cfDNA in patients with diseases is discussed along with the increased cfDNA levels with different tissue damages upon COVID-19 infection. Overall, this article presents important viewpoints and literature review on the potential exploration of cfDNA in Covid management. Here are just minor comments:

  1. Including some conclusion or suggestions line in the abstract would be good to give better context of the flow of ideas in manuscript.
  2. Please cite the main article identifying the mechanism mentioned in Line28.
  3. Please include some detailed figure legends for Fig1, for example, what is happening to nuclear membrane and cellular membrane. Mention NETosis in the figure and/or figure legends for cfDNA release mechanism if applicable or any other mechanism by which cfDNA is released after COVID-19 infection. What is the average size of cfDNA in Kb or Mb? As cfDNA is center of this paper, little bit more information on it would be good in the figure legends and/or its pictorial description.
  4. Is there any report or study exploring high NETosis in patients with obesity, increased age factor etc in Section 2.1. If so, please mention them and cite them.
  5. In Section 2.6, lines 160-162 seem incomplete.
  6. Figure2: “cfDNA increases only few times” should be changed to “cfDNA increases by few folds”; “cfDNA increases a dozen times” should be changed to “cfDNA increases by dozen folds”.

Author Response

 (Reviewer 4)

The authors have reviewed the potential application of cfDNA for COVID-19 diagnostics and management. Role of cfDNA in patients with diseases is discussed along with the increased cfDNA levels with different tissue damages upon COVID-19 infection. Overall, this article presents important viewpoints and literature review on the potential exploration of cfDNA in Covid management. Here are just minor comments:

  1. Including some conclusion or suggestions line in the abstract would be good to give better context of the flow of ideas in manuscript.
  2. Please cite the main article identifying the mechanism mentioned in Line28.
  3. Please include some detailed figure legends for Fig1, for example, what is happening to nuclear membrane and cellular membrane. Mention NETosis in the figure and/or figure legends for cfDNA release mechanism if applicable or any other mechanism by which cfDNA is released after COVID-19 infection. What is the average size of cfDNA in Kb or Mb? As cfDNA is center of this paper, little bit more information on it would be good in the figure legends and/or its pictorial description.
  4. Is there any report or study exploring high NETosis in patients with obesity, increased age factor etc in Section 2.1. If so, please mention them and cite them.
  5. In Section 2.6, lines 160-162 seem incomplete.
  6. Figure2: “cfDNA increases only few times” should be changed to “cfDNA increases by few folds”; “cfDNA increases a dozen times” should be changed to “cfDNA increases by dozen folds”

We would like to express our appreciation for your in-depth comments, suggestions, and corrections, which have greatly improved the manuscript”

Response 1:

We have added the following text to the abstract and introduction.

Purpose has been clarified in abstract Lines 14-17 “We aimed to present the possible mechanisms and sources of cfDNA released during moderate and severe infection. Moreover, we attempt to verify how efficiently cfDNA increase could be applied in COVID-19 risk assessment and how it corresponds with epidemiological data”

and in Introduction in lines 111-115 “The aim of the presented review was to verify how efficiently cfDNA increase could be applied in COVID-19 risk assessment and how it corresponds with epidemiological data. Furthermore, we tried to explain the reasons for the increase in cfDNA during the course of COVID-19 infection, starting from moderate infection, and ending with advanced sepsis and multi-organ failure”

Response 2:

The article has been added: [9] Zhang, H.; Penninger, J.M.; Li, Y.; Zhong, N.; Slutsky, A.S. Angiotensin-converting enzyme 2 (ACE2) as a SARS-CoV-2 receptor: molecular mechanisms and potential therapeutic target. Intensive Care Medicine 2020, 46, 586-590, doi:10.1007/s00134-020-05985-9.

Response 3:

The details for figure one have been added. The following description of the figure has been added.

Lines 30-34 “Analysis of DNA integrity allows to determine the process responsible for the release of cfDNA. Necrosis or NETosis disintegrate the membrane and release random long fragments (<10 000 bp) whereas apoptosis is preceded by shrinkage of the cell, fragmentation into membrane-bound apoptotic bodies, and release of 180-200 bp fragments of DNA (Figure 1) [2].”

Response 4:

Lines 139-148 “MPO-DNA complexes, one of the markers of NETosis, was on a high level in a group of patients with obesity who underwent sleeve gastrectomy compared to healthy controls. Interestingly, patients with reduced MPO-DNA complexes after surgery presented a reduced body weight and BMI and an improved glycemic status. On the other hand, subgroup with persisting high levels of MPO-DNA complexes after surgical treatment had a history of stroke and thromboembolism and therefore may represent a high CV risk population. This may suggest that only weight loss may not modify neutrophil activation [41]. Moreover, numerous of evidence indicate an association of NETs formation in the pathophysiology and complication of diabetes [42-44].”

Response 5: The article has been added [81]  Leung, J.M.; Niikura, M. COVID-19 and COPD. 2020, 56, doi:10.1183/13993003.02108-2020.

The sentence has been redesigned:

“The  most  common  lung  disease,  except  cancer, chronic  obstructive  pulmonary disease (COPD) increases a risk of severe  illness associated with COVID-19.”

Response 6: Figure 2 has been corrected.

Round 2

Reviewer 2 Report

Though many of the recommended changes have been incorporated which have definitely improved the quality of this manuscript, there still remains some areas which require more thought. This reviewer finds the following areas for improvement in this manuscript.

  1. The manuscript needs to be refined with better flow of thought at several instances. For instance. In line 175-179, it is not clear how the included information about the IL6 inhibitors relevant to the point being presented by the authors.
  2. Figure legends need to be improved describing clearly what the figures intend to present. In Figs 1 and 2, it is not clear if the figure is repeated or the authors want to present two similar figures.
  3. The manuscript needs to be thoroughly revised for grammatical mistakes. At several instances the statements are too complicated or confusing. To give a few examples Lines 72-78 do not convey a coherent idea and must be revised.

Author Response

Response to Reviewer

Thank you very much for those constructive comments. According to them, we made point by point following changes in our manuscript:

  1. The manuscript needs to be refined with better flow of thought at several instances. For instance. In line 175-179, it is not clear how the included information about the IL6 inhibitors relevant to the point being presented by the authors.

We have added the following text in lines 194-197(..)”, which limits the development of a hypercytokinemi also called “cytokine storm”. This is a physiological reaction characterized by a sudden release of cytokin in large quantities which might cause multisystem organ failure and even death. (more in “3.1. Blood and immune system” section).”

  1. Figure legends need to be improved describing clearly what the figures intend to present. In Figs 1 and 2, it is not clear if the figure is repeated or the authors want to present two similar figures.

We attempt to clarify this in the previous round of response (lines 297-302) however if this is still blurred, we have added the following figure description: “Schematic representation of how to deal with the patient based on the level of cfDNA reflecting the current condition and the stage of the disease of the COVID-19 patient. Filled spots represent the potential cfDNA releasing organ (details shown in figure 3).”

  1. The manuscript needs to be thoroughly revised for grammatical mistakes. At several instances the statements are too complicated or confusing. To give a few examples Lines 72-78 do not convey a coherent idea and must be revised.

The following paragraph has been reorganized

Line 70-80 “cfDNA plays an essential role in the immune system homeostasis. The cells treated with the native plasma expressed genes whose products maintain immune system homeostasis. Whereas, the cells treated with plasma after DNAse directly elevated production of mRNA for interleukin 8. They also upregulated the complement compounds at the proteomic level [17]. The molecular mechanism of cfDNA is similar to other DAMPs (Damage Associated Molecular Patterns) which can stimulate proinflammatory activity though TLR9 receptor. The principal cytosolic DNA pathway seems to be STING (Stimulator of Interferon Genes) which activates TANK binding kinase 1 (TBK1) and IFN [18]. Besides proinflammatory cfDNA, either in the presence or absence of histones, has been shown to modulate several procoagulant pathways by stimulating thrombosis or inhibits fibrinolytic activity [19,20].”

Moreover, an additional grammatical correction resulted in changes in the following places:

In Line 29, Line 38, Line 51, Line 70, Line 80, Line 107, Line 245, Lines 255-256, Line 302, Line 394, Lines 413-415, Lines 434-435, Lines 461-462.